# Wild worm embryogenesis harbors ubiquitous polygenic modifier variation

**Annalise B Paaby[1]\*, Amelia G White[1], David D Riccardi[1], Kristin C Gunsalus[1,2], Fabio Piano[1,2], Matthew V Rockman[1]\***

[1]Department of Biology and Center for Genomics and Systems Biology, New York University, New York, United States; [2]New York University Abu Dhabi, Abu Dhabi, United Arab Emirates

**Abstract** Embryogenesis is an essential and stereotypic process that nevertheless evolves among species. Its essentiality may favor the accumulation of cryptic genetic variation (CGV) that has no effect in the wild-type but that enhances or suppresses the effects of rare disruptions to gene function. Here, we adapted a classical modifier screen to interrogate the alleles segregating in natural populations of *Caenorhabditis elegans*: we induced gene knockdowns and used quantitative genetic methodology to examine how segregating variants modify the penetrance of embryonic lethality. Each perturbation revealed CGV, indicating that wild-type genomes harbor myriad genetic modifiers that may have little effect individually but which in aggregate can dramatically influence penetrance. Phenotypes were mediated by many modifiers, indicating high polygenicity, but the alleles tend to act very specifically, indicating low pleiotropy. Our findings demonstrate the extent of conditional functionality in complex trait architecture.

**\*For correspondence:** apaaby@nyu.edu (ABP); mrockman@nyu.edu (MVR)

**Competing interests:** The authors declare that no competing interests exist.

## Introduction

The effect of gene disruption on an organism depends on a combination of the gene's function and the genetic background in which it resides (*Chandler et al., 2013*; *Chari and Dworkin, 2013*; *Vu et al., 2015*). The average human genome contains loss-of-function alleles for 100 or more genes, some of which cause known genetic diseases (*Abecasis et al., 2010*; *MacArthur et al., 2012*); disease expression depends on exposure of the disease allele, such as by homozygosity, but also on variants elsewhere in the genome that act as penetrance modifiers (*Hamilton and Yu, 2012*). When looked for, such modifier variation is routinely observed; in model organisms, this phenomenon is recognized as genetic background effects (*Chandler et al., 2013*).

Genetic background effects are an example of cryptic genetic variation (CGV), the class of mutations that affect phenotype under rare conditions (*Gibson and Dworkin, 2004*; *Paaby and Rockman, 2014*). Unlike mutations that are always silent with respect to phenotype, or mutations that always affect phenotype, CGV is invisible until a perturbation changes the molecular, cellular, or developmental processes that govern its phenotypic expression. In addition to genetic perturbations, CGV may be 'released' by environmental exposure, like the modern changes to diet that have been hypothesized to underlie the emergence of complex metabolic diseases in humans (*Gibson, 2009*). The concept of CGV has been of longstanding interest to evolutionary theorists because it explains how populations might store alleles that enable adaptation when conditions change (*Dobzhansky, 1941*; *Waddington, 1956*; *McGuigan et al., 2011*), but its extent, architecture, and role in nature is largely unknown. Most of our empirical knowledge of CGV arises from studies that inhibited the heat shock chaperone protein HSP90 to reveal previously-silent mutational effects across many taxa, which probably represents a general mechanism that buffers genome-wide

**eLife digest** Individuals of the same species have similar, but generally not identical, DNA sequences. This 'genetic variation' is due to random changes in the DNA—known as mutations—that occur among individuals. These mutations may be passed on to these individuals' offspring, who in turn pass them on to their descendants. Some of these mutations may have a positive or negative effect on the ability of the organisms to survive and reproduce, but others may have no effect at all.

The process by which an embryo forms (which is called embryogenesis) follows a precisely controlled series of events. Within the same species, there is genetic variation in the DNA that programs embryogenesis, but it is not clear what effect this variation has on how the embryo develops. Here, Paaby et al. adapted a genetics technique called a 'modifier screen' to study how genetic variation affects the development of a roundworm known as *Caenorhabditis elegans*.

The experiments show that populations of worms harbor a lot of genetic variation that affects how they tolerate the loss of an important gene. One by one, Paaby et al. interrupted the activity of specific genes that embryos need in order to develop. How this affected the embryo, and whether or not it was able to survive, was due in large part to the naturally-occurring genetic variation in other genes in these worms.

Paaby et al.'s findings serve as a reminder that the effect of a mutation depends on other DNA sequences in the organism. In humans, for example, a gene that causes a genetic disease may produce severe symptoms in one patient but mild symptoms in another. Future experiments will reveal the details of how genetic variation affects embryogenesis, which may also provide new insights into how complex processes in animals evolve over time.

functional variation (*Queitsch et al., 2002*; *Yeyati et al., 2007*; *Jarosz and Lindquist, 2010*; *Chen and Wagner, 2012*; *Rohner et al., 2013*).

In this study, we aimed to systematically uncover and characterize genome-wide variation affecting a major metazoan process. *C. elegans* embryogenesis is both complex and typically invariant, which may favor the accumulation of mutations that act in a conditionally-functional manner (*Gibson and Dworkin, 2004*; *Paaby and Rockman, 2014*). We revealed these alleles by perturbing known embryonic genes and measuring differences in penetrance across multiple wild-derived strains.

## Results

To uncover the nature and extent of natural genetic modifiers in *C. elegans* embryogenesis, we individually targeted 29 maternal-effect genes in each of 55 wild strains from around the globe (*Figure 1*). Worms were grown in liquid culture in 96-well plates, and RNAi was delivered by feeding the parental generation *Escherichia coli* expressing dsRNA against the target genes (*Cipriani and Piano, 2011*). Each combination of strain and targeted gene was replicated in at least eight wells, and within each well an average of 10 adult worms contributed hundreds of offspring that were screened as dead or alive. Estimates of embryonic lethality were extracted by the image analysis algorithm DevStaR, which was developed to recognize *C. elegans* developmental stages for this specific application (*White et al., 2013*). We then modeled the probability that an embryo would fail to develop as a function of targeted gene, worm strain, strain-by-gene interaction, and several experimental variables (see 'Materials and methods').

The experiments revealed extensive variation in embryonic lethality caused by genetic differences among strains (*Figure 2*). We observed substantial variation among strains, with some strains exhibiting more embryonic lethality across all targeted genes than other strains, but also significant gene-specific among-strain variation, where particular combinations of gene and strain exhibited surprisingly high or low lethality (*Table 1*). These two classes of variation represent two general mechanisms of modifier action. Informational modifiers (such as suppressors of nonsense mutations in classical screens [e.g., *Hodgkin et al., 1989*], and modifiers of germline RNAi sensitivity in this experiment) alter the effect of the initial perturbation in a non-gene-specific manner, while gene-

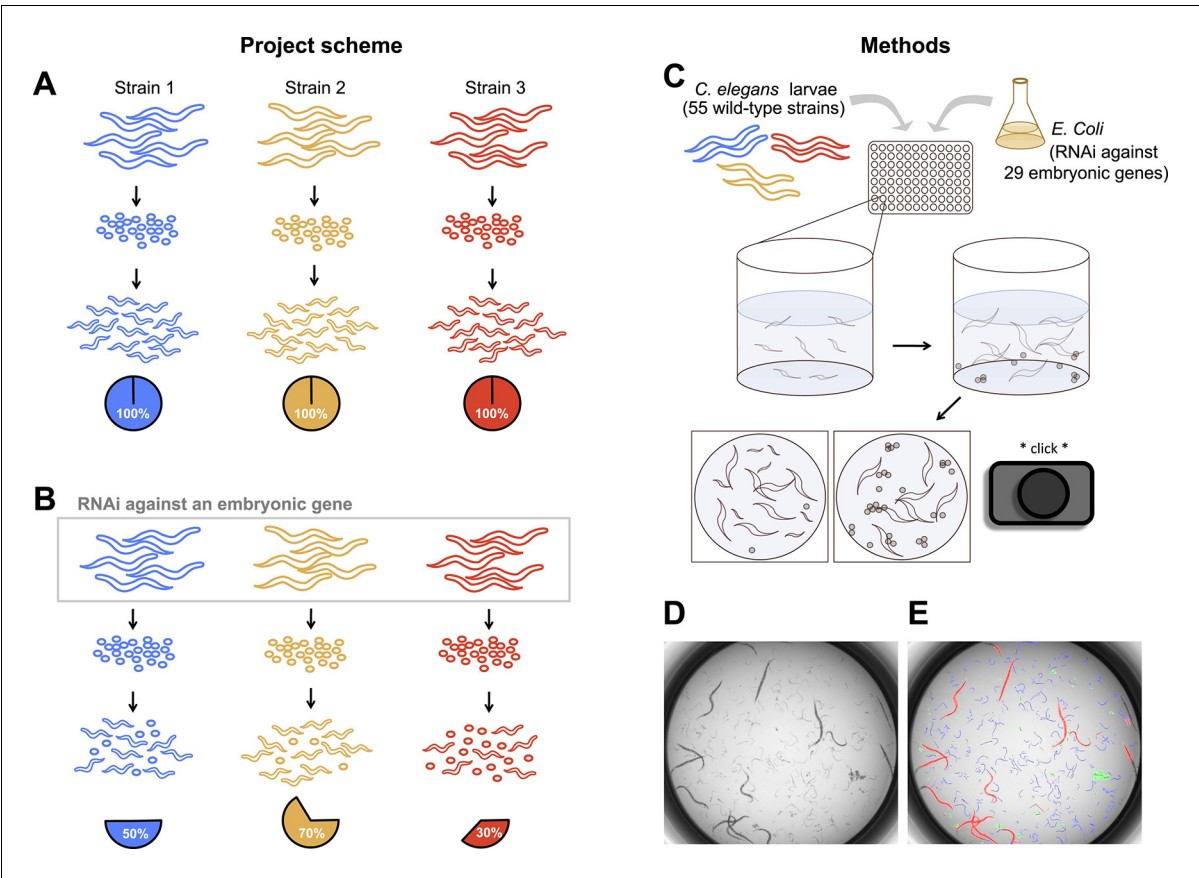

**Figure 1.** Experimental scheme and methods. (**A**) Under ordinary conditions, wild-type *Caenorhabditis elegans* embryos hatch into larvae. (**B**) We targeted maternally-expressed genes by RNAi to induce embryonic lethality that varied in penetrance across strains. (**C**) L1 larvae in the parental generation were fed *Escherichia coli* expressing dsRNA against target genes, in 96-well plates in liquid media. 5 days later, wells were imaged to capture the penetrance of embryonic lethality in the next generation. (**D**, **E**) Raw images were evaluated using DevStaR (**White et al., 2013**), which identified objects as larvae (blue), dead embryos (green), or adults (red).

specific modifiers reveal functional features of the targeted locus. By screening for modifiers of many different perturbations, we are able to quantitatively partition the effects of these mechanisms. Of the variation attributable to heritable modifier variation among worms, half is explained by non-gene-specific informational modifiers and half by gene-specific modifier effects (**Table 1**).

The variation in embryonic lethality attributable to informational modifiers, represented by genetic strain effect in our statistical model, provides an estimate of each strain's sensitivity to exogenous germline RNAi. We observed dramatic variation in sensitivity. Most strains exhibited moderately reduced lethality penetrance relative to the RNAi-sensitive laboratory strain N2, but two strains, the germline RNAi-insensitive strain CB4856 (**Tijsterman et al., 2002**) and the genetically divergent strain QX1211, showed consistently weak penetrance across the targeted genes (**Figure 2**). CB4856 harbors a *ppw-1* loss-of-function mutation that confers resistance to germline RNAi (**Tijsterman et al., 2002**), but sequencing shows that QX1211 and other strains with intermediate sensitivity do not. We found that a *ppw-1* mutation in the N2 background was more sensitive than CB4856, showing high lethality on *mex-3* and *pos-1* (**Figure 2**), indicating that some of the difference between N2 and CB4856 is *ppw-1*-independent. These results demonstrate that insensitivity to germline RNAi is genetically complex and that wild *C. elegans* populations harbor many alleles affecting germline RNAi (**Elvin et al., 2011**; **Pollard and Rockman, 2013**).

Genetic modifiers of RNAi efficacy in our experiment may affect uptake of dsRNA, general RNAi machinery, or tissue-specific RNAi requirements. To distinguish among these, we targeted *tubulin*

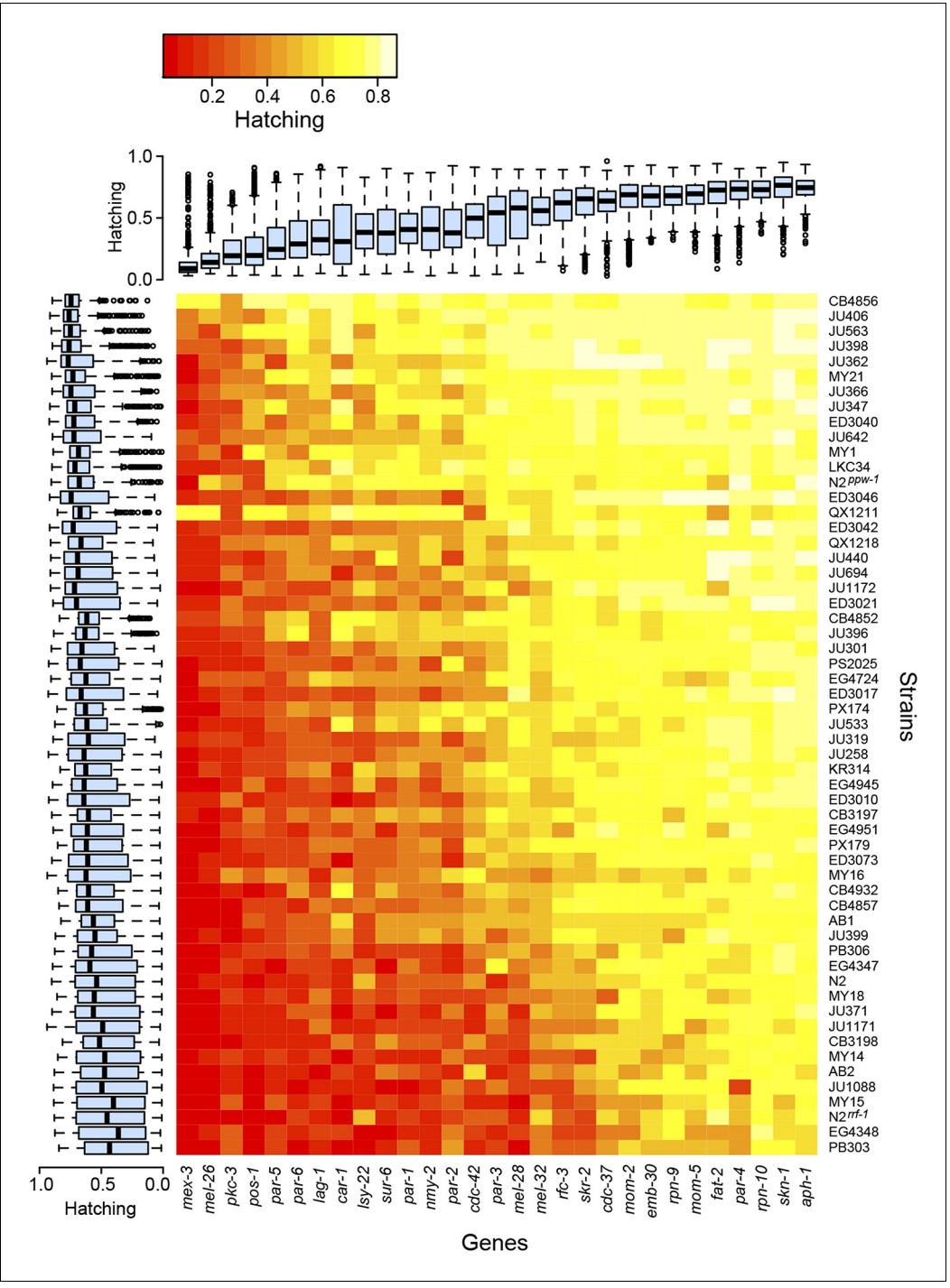

**Figure 2.** Variability in embryonic lethality. Each cell represents the embryonic hatching success for a strain and targeted gene, averaged from at least eight replicate wells. The rows and columns are ordered by average hatching, and boxplots illustrate hatching phenotypes for each strain (across all targeted genes) and for each gene (across all strains).

(*tba-2*), which is expressed ubiquitously. Among wild-type strains, all but four (KR314, JU396, CB4852 and ED3040) showed complete sensitivity to somatic RNAi, indicated by developmental arrest of P$_0$ animals on *tba-2*, which demonstrates that most wild-type strains take up dsRNA and

**Table 1.** Factorial analysis of deviance of lethality phenotypes for 55 wild-type strains in 29 perturbations of germline-expressed genes

|  | DF | Deviance | Resid. DF | Resid. Dev | F | p-value |
|---|---|---|---|---|---|---|
| NULL | – | – | 17,855 | 2,201,873 | – | – |
| Strain | 54 | 338,618 | 17,801 | 1,863,255 | 334.697 | $<10^{-15}$ |
| Targeted gene | 28 | 1,152,310 | 17,773 | 710,945 | 2196.584 | $<10^{-15}$ |
| Adults per well | 1 | 35,318 | 17,772 | 675,627 | 1885.113 | $<10^{-15}$ |
| Date | 1 | 2406 | 17,771 | 673,221 | 128.416 | $<10^{-15}$ |
| Strain × gene | 1512 | 349,415 | 16,259 | 323,806 | 12.334 | $<10^{-15}$ |
| Strain × adults per well | 54 | 6715 | 16,205 | 317,091 | 6.637 | $<10^{-15}$ |
| Gene × adults per well | 28 | 7358 | 16,177 | 309,732 | 14.026 | $<10^{-15}$ |

The table rows report information associated with each term in our statistical model (see 'Materials and methods'), which represent distinct sources for the variation we observed in embryonic lethality. All terms were highly significant, including the strain-by-gene interaction, which represents variation attributable to cryptic genetic modifiers that act gene-specifically. This term and the strain term, which represents variation attributable to informational modifiers affecting germline RNAi, explain similar amounts of variation, and together account for 31% of the total deviance.

are capable of RNAi. An *rrf-1* deletion mutant, which is sensitive to RNAi against genes expressed in the germline but resistant to RNAi in most somatic tissues (*Yigit et al., 2006*; *Kumsta and Hansen, 2012*), grew to adulthood but laid dead embryos, suggesting that germline RNAi successfully silenced maternal *tba-2* required for embryonic development. The four somatically-resistant wild strains also exhibited embryonic lethality on *tba-2* and other germline-expressed genes, confirming that the modifier variability acts tissue-specifically.

Gene-specific modifiers explain as much of the total variation as the informational modifiers, as estimated by the strain-by-gene interaction term in our model (*Table 1*), and represent cryptic genetic variation in developmental processes. The modifiers could act via network bypasses, where loss of the targeted gene reveals variation among strains in developmental network structure (e.g., *Zhang and Emmons, 2000*). Gene-specific modifiers could also act on the extent of the knockdown at a gene-specific level, in a manner akin to intragenic suppressors, resulting in variable residual activity of the targeted gene. This latter class potentially includes gene-specific variation in RNAi sensitivity, perhaps due to heritable variation in transcriptional licensing (*Shirayama et al., 2012*; *Seth et al., 2013*), and variation in wild-type expression level of the targeted gene, due to cis- or trans-acting regulatory variation.

Each of the 29 genes we targeted showed significant strain-by-gene interaction coefficients, indicating that genetic modifiers of embryonic gene perturbations are pervasive in natural populations. The coefficients, which are statistical estimates of the gene-specific cryptic phenotypes (see 'Materials and methods'), exhibit low correlations between gene perturbations known to share function: 36 gene pairs have known physical or genetic interactions, but these did not show significantly elevated phenotypic correlations ($\chi^2 = 2.30$, df = 1, p = 0.13). For example, despite high interaction within the *par* network, which underlies polarization of the zygote, the average pairwise *par* gene correlation was no higher than the average correlation across all genes (*Supplementary file 1*). Coefficients for *par-3* and *par-6* were correlated (correlation = 0.40, p = 0.003), but not for *par-3* and *pkc-3* (correlation = −0.17, p = 0.24) or *par-6* and *pkc-3* (correlation = 0.12, p = 0.41), even though their proteins together comprise the anterior polarity complex (*Munro et al., 2004*). This indicates that the cryptic genetic modifiers have low developmental pleiotropy (*Paaby and Rockman, 2013*). That is, variation at these loci must influence a very restricted suite of developmental events, since only specific perturbations uncover evidence of their phenotypic effects. For those associated with polarization of the zygote, this may be explained by the high degree of redundancy observed in the process (*Beatty et al., 2010*; *Fievet et al., 2013*; *Motegi and Seydoux, 2013*), as redundancy allows shared function of some factors and specificity of others. Exceptions to the overall trend of low correlation between gene perturbations are discussed below, in the context of genome-wide associations. The

low pleiotropy of cryptic alleles may be a result of purifying selection, which over evolutionary time should deplete populations of pleiotropic alleles as they may be more likely to be deleterious (**Stern, 2000**).

Our quantitative-genetic approach is uniquely able to discern modifier effects that depend simultaneously on variants at many loci. In order to evaluate the polygenicity of the gene-specific variation we observed, and to ask whether cryptic alleles are rare or common in populations, we assessed whether genome-wide genetic similarity among strains explained patterns of phenotypic similarity (**Kang et al., 2008**). Specifically, we estimated the genomic heritability of the strain-by-gene coefficients. This approach estimates the proportion of gene-specific modifier effects caused by alleles of intermediate frequency at many loci, as these are best captured in estimates of strain relatedness.

We found that for most of the perturbations, variation in lethality penetrance is due to common alleles at many contributing cryptic loci. Of the 29 genes we targeted, 12 exhibited gene-specific modifier variation with genomic heritability estimates greater than 0.80; for 19 genes, estimates were greater than 0.60 (**Table 2**). However, genotypic similarity failed to explain phenotypic similarity for perturbations of *emb-30*, *mel-32*, *mex-3*, *mom-5*, *par-3* and *sur-6* (**Table 2**). Because these genes exhibit nonzero variance in their associated strain-by-gene interaction coefficients, the strains necessarily harbor cryptic genetic differences affecting lethality under these perturbations. Thus, the genetic architecture of the cryptic variation associated with these genes is likely comprised of few loci, rarer alleles, or both.

To locate genome regions harboring gene-specific modifiers, we performed genome-wide association (GWA) mapping using the strain-by-gene interaction coefficients as phenotypes. GWA in *C. elegans* benefits from high linkage disequilibrium in this species, which reduces the number of tests required to scan the genome, and from high biological replication, which reduces the number of required genotypes relative to human GWA (**Rockman and Kruglyak, 2009**; **Andersen et al., 2012**). Nine of the 29 analyses identified at least one single nucleotide polymorphism (SNP) associated with phenotype under a strict Bonferroni-corrected threshold for significance (**Supplementary file 2**). Across all tests, a total of 19 SNPs or SNP haplotype blocks, defined by SNPs in high linkage disequilibrium ($R^2 > 0.9$), exhibited significant associations at that threshold (**Supplementary file 2**), while many additional variants exhibit suggestive associations (p < 0.001).

To validate the GWA findings, we introgressed a segment of chromosome II from strain N2 into the genome of wild isolate EG4348. Gene-specific modifier phenotypes for *lsy-22* and *pkc-3* both have suggestive associations with SNPs on the right arm of chromosome II (the SNPs for *lsy-22* are independent of those for *pkc-3* [$R^2 = 0.03$], which reside approximately a megabase away, implicating distinct cryptic modifiers). N2 exhibits low lethality when *lsy-22* is targeted but high lethality on *pkc-3*, and

**Table 2.** Genome heritability estimates for CGV phenotypes associated with 29 targeted genes

| Targeted gene | Heritability estimate | p-value |
|---|---|---|
| *aph-1* | 0.6747 | 0.16 |
| *car-1* | 0.9149 | 0.02 |
| *cdc-37* | 0.7308 | 0.11 |
| *cdc-42* | 0.3639 | 0.29 |
| *emb-30* | 0.0000 | 0.46 |
| *fat-2* | 0.3548 | 0.32 |
| *lag-1* | 0.9075 | 0.01 |
| *lsy-22* | 0.1270 | 0.43 |
| *mel-26* | 0.8245 | 0.05 |
| *mel-28* | 0.8410 | 0.04 |
| *mel-32* | 0.0000 | 0.47 |
| *mex-3* | 0.0000 | 0.76 |
| *mom-2* | 0.7485 | 0.09 |
| *mom-5* | 0.0000 | 0.46 |
| *nmy-2* | 0.4841 | 0.26 |
| *par-1* | 0.7871 | 0.08 |
| *par-2* | 0.9719 | 0.01 |
| *par-3* | 0.0000 | 0.77 |
| *par-4* | 0.9032 | 0.07 |
| *par-5* | 0.6640 | 0.15 |
| *par-6* | 0.9258 | 0.01 |
| *pkc-3* | 0.8136 | 0.06 |
| *pos-1* | 0.7307 | 0.10 |
| *rfc-3* | 0.6958 | 0.13 |
| *rpn-9* | 0.8715 | 0.02 |
| *rpn-10* | 0.8397 | 0.05 |
| *skn-1* | 0.8599 | 0.03 |
| *skr-2* | 0.8961 | 0.02 |
| *sur-6* | 0.0000 | 0.47 |

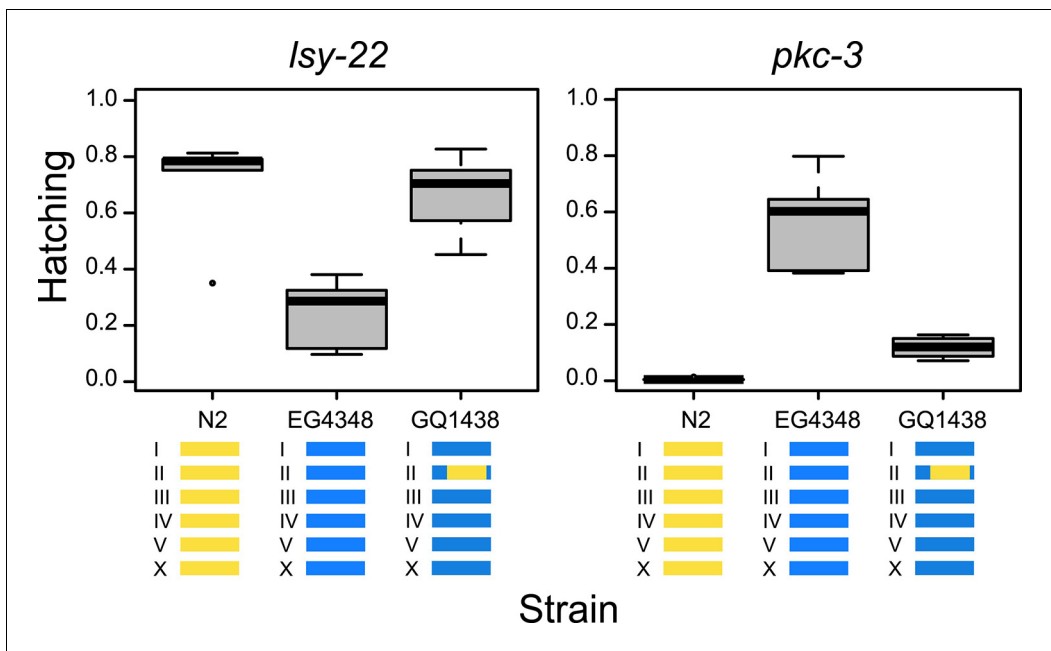

**Figure 3.** Tests for gene-specific modifiers. Introgression of part of chromosome II from strain N2 (yellow) into strain EG4348 (blue) rescues the N2 phenotype on *lsy-22* (F = 12.15, DF = 2, p = 0.001) and *pkc-3* (F = 55.87, DF = 2, p < 0.001); genome-wide analyses found associations between this region and hatching phenotypes for both *lsy-22* and *pkc-3*.

The following source data is available for figure 3:

**Source data 1.** This file provides source data for *Figure 3*, which reports hatching for three different strains targeted by RNAi against genes lsy-22 and pkc-3.

EG4348 shows the opposite pattern; in both comparisons, the introgression rescued the original N2 phenotype (*Figure 3*). These results demonstrate that cryptic variants within the introgression modify the effects of *lsy-22* and *pkc-3* perturbations.

To distinguish between intragenic and extragenic modifiers, we considered the list of 129 associated SNPs (in 27 haplotype blocks) with p-values less than $10^{-4}$ (*Supplementary file 2*), all of which exceed the significance of the validated *lsy-22* and *pkc-3* modifiers. These associations were spread across 15 targeted-gene phenotypes. No SNPs lie within or near the locus of the targeted gene, with the exception of one SNP within the *mel-28* locus that associates with the *mel-28* phenotype. The *mel-28* phenotype is also associated with multiple other SNPs elsewhere in the genome. Thus, most of the CGV detectable by GWA is caused by extragenic modifiers.

Extragenic modifiers may work by affecting, in trans, the expression level of the targeted gene. Recent work shows that differences in severity of RNAi phenotype, for four *C. elegans* strains perturbed at electron transport chain genes, are associated with differences in expression level of the targeted gene (*Vu et al., 2015*). However, we find no evidence for the reported pattern of lower expression explaining more severe phenotypes. We examined published transcript abundances for our 29 target genes measured in 4-cell embryos (*Grishkevich et al., 2012*) under standard conditions in five strains. Five of the genes exhibited significant variation in expression among the strains. In contrast, RNAi against 28 induced significant gene-specific variation in embryonic lethality among the five strains. Overall, both for genes with significant variation and for the whole set, lower expression of the target gene was usually correlated with less severe RNAi phenotypes (20 of 29 genes, p = 0.06), though the correlations are weak. Although undetectable differences in transcript level may nevertheless contribute to embryonic survival, these results suggest that much of the gene-specific modifier effect we observe depends on variation beyond the target gene.

Our GWA mapping identified few SNPs associated with more than one phenotype. For example, lethality phenotypes for 4 of the 7 targeted polarity genes (*par-2, -4, -6* and *pkc-3*) were associated with SNPs, but none were shared. The discrete nature of the genotype–phenotype associations further implies low developmental pleiotropy of the cryptic alleles; variants with effects under one perturbation have no detectable effects under another.

However, the rare instances of multiple associations for individual SNPs implicate a relationship between the targeted genes (*Supplementary file 2*). The co-association of SNPs in a haplotype block on chromosome IV with lethality phenotypes for *rpn-9* and *rpn-10* support a known relationship, as *rpn-9* and *rpn-10* both encode non-ATPase regulatory subunits of the proteasome and are predicted to interact with each other (*Zhong and Sternberg, 2006*; *Lee et al., 2008*). The haplotype, which spans approximately 10 kb, was also significantly associated with lethality phenotypes for *car-1, mom-5*, and *skn-1*; *skn-1* has a role in proteasome-mediated protein homeostasis (*Li et al., 2011*). Separately, modifier phenotypes for *pkc-3*, involved in anterior-posterior polarity in the early embryo, and *rfc-3*, which shows homology to DNA replication factors C and effects on cell cycle synchrony (*Piano et al., 2002*), are associated with SNPs on both chromosome III and X. Because the co-associations occur twice, with unlinked SNPs ($R^2 = 0.26$), they implicate the presence of at least two interacting cryptic alleles and provide independent lines of evidence for a relationship between *pkc-3* and *rfc-3*, genes with no reported interactions or shared functions.

## Discussion

We have uncovered pervasive CGV that modifies the probability that an embryo will survive a gene perturbation. By evaluating the effects of naturally-occurring mutations on gene knockdowns, we explored a genotypic space that is distinct from that accessible to conventional screens. Our findings provide complementary insight, including discovery of modifier activity that may be detectable only when effects are moderate (*Fievet et al., 2013*) or polygenic (*Mackay, 2014*).

We describe the variation we uncovered as 'cryptic' because its effect on embryonic survival is dramatically magnified under perturbed conditions. Without gene perturbation, our strains exhibit little embryonic lethality. However, under ordinary conditions the strains vary in gene expression and other cellular or developmental phenotypes (*Grishkevich et al., 2012*; *Farhadifar et al., 2015*), which may be the mechanisms by which the cryptic alleles influence the penetrance of the primary perturbation. Previously, we and others have described such differences as variation in 'intermediate' phenotypes (*Félix and Wagner, 2008*; *Paaby and Rockman, 2014*); whether a genetic variant is cryptic requires definition of the focal phenotype, since even at the morphological level an allele can be cryptic in one trait but penetrant in another (*Duveau and Félix, 2012*).

Exploration of CGV is not new: CGV has been demonstrated following perturbation of candidate genes (*Gibson and Hogness, 1996*; *Dworkin et al., 2003*; *Cassidy et al., 2013*; *Chandler et al., 2013*; *Chari and Dworkin, 2013*); its potential role in adaptive evolution has been considered in diverse systems (*Dobzhansky, 1941*; *Waddington, 1953*; *Masel, 2006*; *Ledon-Rettig et al., 2010*; *McGuigan et al., 2011*; *Duveau and Félix, 2012*; *Rohner et al., 2013*); and most extensively, it has been characterized following inhibition of HSP90 (*Rutherford and Lindquist, 1998*; *Queitsch et al., 2002*; *Yeyati et al., 2007*; *Jarosz and Lindquist, 2010*). Here, we show by systematic evaluation that the phenomenon of conditionally functional variation pervades even the highly stereotyped and controlled process of embryogenesis.

We found that gene-specific cryptic variation affects every targeted gene, implying that wild populations harbor many enhancers and suppressors of critical embryonic genes. In humans, such penetrance modifiers may mediate expression of genetic diseases arising from loss-of-function mutations (*Abecasis et al., 2010*; *Hamilton and Yu, 2012*; *MacArthur et al., 2012*), and if their crypsis is environmentally influenced they may also explain modern disease susceptibility (*Gibson, 2009*). Our screen also revealed dramatic variation among wild-type strains in their responses to exogenous RNAi in the germline. Somatic RNAi response has been shown to influence *C. elegans* susceptibility to viral infection; variation in germline RNAi may affect vertical viral transmissibility (*Félix et al., 2011*) as well as transposon activity (*Sijen and Plasterk, 2003*; *Vastenhouw and Plasterk, 2004*). The variation we describe illustrates how conditionally-functional relationships between genes may pervade the variation on which natural selection acts, affecting how complex traits evolve (*True and Haag, 2001*; *Félix, 2007*; *Wang and Sommer, 2011*; *Verster et al., 2014*) and the nature of their

genetic architecture (*Mackay, 2014*). Moreover, this variation has major implications for model system biologists that work with a single genetic strain.

## Materials and methods

### *C. elegans* strains

We evaluated laboratory strain N2, originally derived from Bristol, England, and 54 wild-type strains derived from populations around the world. The wild-type strains were chosen with reference to genotype data (*Rockman and Kruglyak, 2009*; *Andersen et al., 2012*); we avoided haplotype-identical isolates, which can occur even across disparate sampling locations, and included the most diverged genotypes at the population level. The wild-type strains were: AB1, AB2 (Adelaide, Australia), CB3197, PS2025 (Altadena, CA, USA), CB3198 (Pasadena, CA, USA), CB4852 (Rothamsted, England), CB4856 (Hawaii, USA), CB4857 (Claremont, CA, USA), CB4932 (Taunton, England), ED3010, ED3017, ED3021 (Edinburgh, Scotland), ED3040 (Johannesburg, South Africa), ED3042, ED3046 (Western Cape, South Africa), ED3073 (Limuru, Kenya), EG4347, EG4348, EG4945, EG4951 (Salt Lake City, UT, USA), EG4724 (Amares, Portugal), JU1088 (Japan), JU1171, JU1172 (Chile), JU258 (Madeira, Portugal), JU301 (LeBlanc, France), JU319, JU347 (Merlet, France), JU362, JU366, JU371, JU694 (Franconville, France), JU396, JU398, JU399, JU406 (Hermanville, France), JU440 (Beauchene, France), JU533 (Primel, France), JU563 (Sainte Barbe, France), JU642 (Le Perreux, France), KR314 (Vancouver, Canada), LKC34 (Madagascar), MY1 (Lingen, Germany), MY14, MY15, MY16 (Mecklenbeck, Germany), MY18, MY21 (Roxel, Germany), PB303 (isolated from an isopod from Ward's Biological Supply), PB306 (isolated from an isopod from Connecticut Valley Biological Supply), PX174 (Lincoln City, OR, USA), PX179 (Eugene, OR, USA), QX1211 (San Francisco, CA, USA), and QX1218 (Berkeley, CA, USA). Isolates were acquired from the Caenorhabditis Genetics Center or kindly shared by members in the worm community. We also assayed N2 mutants NL2557, which carries a deletion at *ppw-1* (*pk1425*) that confers resistance to RNAi in the germline (*Tijsterman et al., 2002*), and NL2098, which carries a deletion at *rrf-1* (*pk1417*) that confers resistance to RNAi in most somatic tissues (*Yigit et al., 2006*; *Kumsta and Hansen, 2012*). These were provided by the Caenorhabditis Genetics Center, which is funded by NIH Office of Research Infrastructure Programs (P40 OD010440).

### Phenotyping embryonic lethality in liquid culture

Worms were grown to large numbers on agarose-media plates, and healthy embryos at least two generations past starvation or thawing were collected using standard bleaching techniques. For each strain, ~10,000 embryos were plated onto a 15 cm agarose-media plate densely seeded with *E. coli* OP50. Worms were reared at 20°C with food until gravid, then bleached and the embryos synchronized to the arrested L1 larval stage by rocking in M9 buffer overnight at 20°C. Following the methodology for growing and imaging worms in 96-well plates described in ref. 27, larvae were washed and diluted to 10 worms per 20 µl of S buffer with additives. Worms were dispensed with a peristaltic pump (Matrix Wellmate) in 20 µl volumes into wells of flat-bottomed 96-well plates (in rows, 8 strains per plate) already containing 30 µl of the appropriate RNAi feeding bacteria. Each plate was replicated eight times, and N2 was dispensed on every plate. After dispensing, plates were stored at 20°C in sealed humid chambers for 5 days. Three sets of eight worm strains were dispensed per experimental cycle; we performed a total of three cycles over 3 months.

### RNAi vectors

In our initial survey, we targeted 41 germline-expressed genes and one somatic gene (*tba-2*). The germline-expressed genes were chosen following exploratory examination of a larger set of embryonic genes for which observations of embryonic lethality phenotypes were reported on wormbase. org. The final set of 41 were selected by eliminating genes with effects on post-embryonic development or sterility, by including genes with a range of lethality penetrance in N2, and by including the seven core members of the *par* pathway. We targeted the genes by feeding the worms HT115 *E. coli* bacteria expressing dsRNA for their targets. Bacteria had been transformed with pL4440-derived RNAi feeding vectors into which target DNA had been cloned (*Timmons et al., 2001*) and which carry genes for ampicillin and tetracycline resistance. We also included *E. coli* carrying the empty

pL4440 vector, for a total of 43 RNAi vectors in the survey. The majority of the RNAi vectors we used were obtained from the Ahringer feeding library (*Kamath and Ahringer, 2003*). These included: *aph-1* (VF36H2L.1), *car-1* (Y18D10A.17), *cdc-37* (W08F4.8), *cdc-42* (R07G3.1), *ceh-18* (ZC64.3), *cyb-2.1* (Y43E12A.1), *emb-30* (F54C8.3), *fat-2* (W02A2.1), *gad-1* (T05H4.14), *lag-1* (K08B4.1), *lin-5* (T09A5.10), *lsy-22* (F27D4.2), *mel-26* (ZK858.4), *mel-28* (C38D4.3), *mel-32* (C05D11.11), *mes-1* (F54F7.5), *mex-3* (F53G12.5), *mom-2* (F38E1.7), *mom-5* (T23D8.1), *nmy-2* (F20G4.3), *nos-3* (Y53C12B.3), *ooc-3* (B0334.11), *par-1* (H39E23.1), *par-2* (F58B6.3), *par-3* (F54E7.3), *par-5* (M117.2), *par-6* (T26E3.3), *pkc-3* (F09E5.1), *pos-1* (F52E1.1), *rfc-3* (C39E9.13), *rpn-10* (B0205.3), *rpn-12* (ZK20.5), *rpn-9* (T06D8.8), *skn-1* (T19E7.2), *skr-2* (F46A9.4), *spat-1* (F57C2.6), *spat-2* (Y48A6B.13), *sur-6* (F26E4.1), *tba-2* (C47B2.3), and *ztf-1* (F54F2.5). We also used two feeding vectors created and kindly shared by M. Mana, for genes *gpb-1* (F13D12.7) and *par-4* (Y59A8B.14).

We constructed a frozen RNAi bacterial feeding library in 96-well plates with 20% glycerol. The bacteria were distributed across the plates in columns (12 vectors per plate); the *mom-2* vector was included on every plate. Using a 96-pin replicator, bacterial colonies were transferred from the frozen libraries and grown on LB agar plates (100 µg/ml ampicillin, 12 µg/ml tetracyclin). LB broth (50 µg/ml ampicillin) in 96-deep-well plates was inoculated from the solid cultures using the pin replicator and grown overnight in a 37°C shaker. Cultures were induced with 1 mM IPTG for two hours and dispensed into 96-well flat-bottom plates using a Tecan Aquarius robot.

## Excluding genes from the analysis

Although we evaluated 41 genes in our experiment, in our final analysis we included results only for 29. Perturbing *gpb-1* and *lin-5* induced growth defects in multiple strains such that the parental generation of worms failed to develop to reproductive maturity, indicating that these genes have effects outside of embryogenesis. We also identified ten genes (*ceh-18*, *cyb-2.1*, *gad-1*, *mes-1*, *ooc-3*, *nos-3*, *rpn-12*, *spat-1*, *spat-2* and *ztf-1*) that induced no or extremely low embryonic lethality. As they were indistinguishable from the empty vector negative control, we excluded them from analysis.

## Image acquisition and data extraction

5 days after the experimental cycle was initiated, the L1 larvae had developed into egg-laying adults and consumed the RNAi bacteria so that the wells were optically clear. Wells were photographed at the point at which viable embryos had hatched but not developed past early larval stages. We captured single images of each well using a DFC340 FX camera and a Z16 dissecting microscope (Leica Microsystems, Inc., Buffalo Grove, IL), a Bio-precision motorized stage with adaptors for the 96-well plates and stage fittings (Ludl, Inc., Hawthorne, NY), and Surveyor software from Media Cybernetics, Inc. (Warrendale, PA). We used a 1.2 ms exposure at 17.3× magnification.

Data were extracted from the images using the automated image analysis system DevStaR (*White et al., 2013*). DevStaR is an object recognition machine that classifies each object in the image as an adult, larva or embryo using a support vector machine and global shape recognition. Embryonic lethality estimates were derived from the proportion of embryos in each well relative to all progeny (embryos plus larvae). During the development of DevStaR, each of the approximately 30,000 images in this experiment were manually evaluated and assigned qualitative scores for the number of embryos and the number of larvae, and exact counts were determined for the adults in each well. These data provided independent phenotype estimates and demonstrate that DevStaR is accurate and reliable (*White et al., 2013*), and we used the manually-collected adult count data in our analyses evaluating the number of adults in each well.

## Statistical analyses

The counts of dead embryos and living larvae from each experimental well were bound together as a single response variable and modeled using a generalized linear model with a quasi-binomial error structure. In the central analysis, in which we evaluated 55 strains and 29 genes, the model included main effects of strain, targeted gene, number of adult worms per well, and experimental date; and interaction terms for strain-by-gene, strain-by-adults and gene-by-adults, in the form:

$$E(Y) = g^{-1}(\beta_0 + \beta_{Strain}X_{Strain} + \beta_{Gene}X_{Gene} + \beta_{Adults}X_{Adults} + \beta_{Date}X_{Date}$$

$$+ \beta_{Strain*Gene}X_{Strain}X_{Gene} + \beta_{Strain*Adults}X_{Strain}X_{Adults} + \beta_{Gene*Adults}X_{Gene}X_{Adults})$$

where $g^{-1}$ represents a logit link function. The analysis was conducted using the *glm* function in *R Development Core Team (2010)* and model fit was examined with the deviance statistic.

Coefficients from the strain-by-gene interaction term in this model were used as estimates of gene-specific CGV, as they provide quantitative measures of probability of embryonic lethality associated with each perturbation after accounting for contributions from the general degree of lethality of the perturbation, the strain effect associated with variation in informational modifiers affecting germline RNAi, and other experimental variables. The significance of each coefficient was computed by assessing the coefficient ratio against the *t*-distribution using the *summary.glm* function. We also performed a mixed-model analysis using the *glmer* function in the R package *lme4* (*Bates, 2010*) with a logit link function and a binomial error structure, in which all effects except the number of adults were specified as random. Results from this analysis were consistent with the fixed-effects analysis, including tight correlation between the fixed-effect coefficients and the mixed-effect estimates and between the downstream GWAS results; we only report results from the fixed-effects analysis. Other analyses, including those exploring confounding effects of experimental design, fitted models with additional terms for well position and bacterial source to subsets of the data. To identify best-fitting models, terms were sequentially reduced from the full model and model comparison was achieved with the F statistic.

Correlations among gene perturbations were estimated using the Spearman Rank method in R. The coefficients, extracted from the generalized linear model, for each strain on each targeted gene were compared for each pairwise combination of genes. Evidence for known interactions among pairs of genes was collated from wormbase.org (February 2015) and includes physical and genetic interactions. We tested whether gene pairs with known interactions had higher phenotypic correlations using the Kruskal–Wallis method in R.

## Experimental replication and controls

Because we arranged worm strains in fixed rows and RNAi vectors in fixed columns across the 96-well experimental plates, well position was a potentially confounding source of variation in the data. The source of each bacterial culture was also potentially confounding, as each culture was grown independently for each strain on a plate. To estimate the contribution of these variables to the lethality phenotypes, we examined hatching variation for strain N2 on targeted gene *mom-2*, which we included in every plate. The dataset includes counts of dead and alive offspring from 285 experimental wells. Independent cultures of *E. coli* bacteria expressing dsRNA against *mom-2* only weakly affected hatching (F = 3.12, DF = 2, p = 0.046) (*Table 3*), and whether a well was on the edge, near the edge, or in the center of the plate had no effect on phenotype (F = 1.39, DF = 2, p = 0.251).

With the exception of N2, strains were assayed in one of three date batches. To evaluate the relative importance of date, we examined the N2 lethality phenotypes for all 29 lethality-inducing genes across the three dates. While the date effect was statistically significant, it explained only 1.9% of the deviance; the gene effect explained 86.6% of the deviance (*Table 4*). The model that best fits the data also includes main and interaction terms for the number of adults per well, but their effects are similarly negligible.

**Table 3.** Factorial analysis of deviance of strain N2 lethality on targeted gene *mom-2*

|  | Df | Deviance | Resid. Df | Resid. Dev | F | Pr (>F) |
|---|---|---|---|---|---|---|
| NULL | – | – | 284 | 9191.7 | – | – |
| Date | 1 | 1060.26 | 283 | 8131.4 | 38.397 | $2.0 \times 10^{-09}$ |
| Bacterial source | 2 | 172.47 | 281 | 7958.9 | 3.123 | 0.04556 |

**Table 4.** Factorial analysis of deviance of strain N2 lethality phenotypes across 29 targeted genes

|  | Df | Deviance | Resid. Df | Resid. Dev | F | Pr (>F) |
|---|---|---|---|---|---|---|
| NULL | – | – | 2254 | 280,706 | – | – |
| Silenced gene | 28 | 221,081 | 2226 | 59,624 | 378.3275 | $<2 \times 10^{-16}$ |
| Date | 1 | 6090 | 2225 | 53,534 | 291.8186 | $<2 \times 10^{-16}$ |
| Adults per well | 1 | 249 | 2224 | 53,285 | 11.9248 | 0.00056 |
| Silenced gene × date | 28 | 5265 | 2196 | 48,020 | 9.0099 | $<2 \times 10^{-16}$ |
| Silenced gene × adults per well | 28 | 2423 | 2168 | 45,597 | 4.1467 | $2.7 \times 10^{-12}$ |

## Genome-wide association tests

Association analyses of the gene-specific embryonic lethality phenotypes were implemented with the *emma.ML.LRT* function in the R package *emma*, which controls for population structure using a kinship matrix and performs efficient mixed-model association using maximum likelihood (*Kang et al., 2008*). The kinship matrix was determined from a total of 41,188 SNPs across 53 strains; we excluded strains CB4856 and QX1211, as they are essentially insensitive to RNAi in the germline. The SNP genotypes are as described in *Andersen et al. (2012)* and were downloaded from the website of E Andersen (http://groups.molbiosci.northwestern.edu/andersen/Data.html). We assayed six wild isolates not fully genotyped by that study; see our imputation method below. The phenotype values were the coefficients estimated from the strain-by-gene interaction by the generalized linear model, as they include strain contributions to lethality minus the strain effect, the date effect, and other effects of experimental design. We evaluated SNPs with minor allele counts of 6 or more, which allowed us to interrogate 9362 SNPs. Of these, 3057 exhibit unique genotype identities across the 53 strains, and the strict threshold for significance, following Bonferroni correction for multiple tests, was determined at 0.05/3,057, or $1.6 \times 10^{-5}$. Genomic heritability estimates for each of the cryptic phenotypes represented by the strain-by-gene coefficients was determined from the genetic and residual error variance components estimated by restricted maximum likelihood, using the function *emma.REMLE*. Significance was tested by 1000 permutations of strain phenotypes.

## Genotype imputation

Six wild isolates in our study were not fully genotyped using the RAD-seq method by *Andersen et al. (2012)*, and we used the following procedure to impute genotypes at the full set of SNPs. If the strain was identical at the 1454 SNPs assayed by *Rockman and Kruglyak (2009)* to a strain genotyped by RAD-seq, we used the RAD-seq data of the matching strain. This allowed us to use genotype data of CB4854 for CB3197; JU310 for JU301; JU311 for JU319; JU367 for JU371; and MY10 for MY21. In each of these cases, multiple RAD-sequenced strains collapse into groups of strains that are also identical at the 1454 SNPs, suggesting that this procedure is reliable. Only in the case of JU366 do we encounter uncertainty. At the 1454 SNP markers, this strain is identical to JU360, JU363, and JU368 (and three others not RAD-sequenced). JU360 and JU368 have identical RAD-seq haplotypes, but JU363 is different at 224 sites (of which 164 were tested for association with phenotype). We substituted both JU360 and JU363 as proxies for JU366 and ran the full GWAS pipeline twice; the differences in outcome were negligible, with extremely tight correlation among SNP p-values across all tests and no differences in the set of statistically significant SNPs.

## Validation of CGV by introgression

We created the strain QG611, which carries two markers (*mIs12*, expressing GFP in the pharynx, and *juIs76*, expressing GFP in the motor neurons) in the N2 wild-type background. The markers are positioned at the approximate middle and right end of chromosome II, respectively (precise locations are unknown), which flank the region for which *lsy-22* and *pkc-3* phenotypes were associated. We crossed QG611 to wild-type strain EG4348 and then backcrossed to EG4348 for 20 generations, retaining the N2 introgression by selecting for the double markers. The introgression strain, QG1438, carries the N2 haplotype from approximately II 3,174,000 to the right of II 14,430,751. To

test the effect of the introgression on *lsy-22* and *pkc-3* perturbations, RNAi was induced by feeding on agarose plates following standard protocols (wormbook.org): test worms were singled onto plates, 6 replicates each, at the L4 stage following bleaching and developmental synchronization; worms were transferred daily for 3 days and the number of dead embryos and hatched larvae were counted 24 hr after transfer. Test strains included QG611 (the GFP constructs in QG611 have no effect on phenotype relative to N2, data not shown), EG4348, and QG1438. The data were analyzed using a generalized linear model with a quasi-binomial error structure to test the effect of strain on embryonic lethality.

## Genome sequencing and off-target predictions

Seventeen strains (AB1, AB2, CB3198, CB4852, CB4856, EG4347, EG4348, JU319, JU371, JU1088, JU1171, MY1, MY16, MY18, PB306, PX174, PX179) were examined for sequence variation at the RNAi target sites. Sequences were derived from 100-bp paired-end reads run on an Illumina HiSeq 2500 that were mapped to the N2 reference (ce10) using *stampy* (*Lunter and Goodson, 2011*) and variant-called with *samtools* (*Li et al., 2009*). We observed nucleotide variation in these genes, but zero mutations in the exons targeted by the RNAi clones we used. Thus, we exclude RNAi mismatch via target locus sequence variation as a source of the phenotypic variation we observed. Off-target predictions for our RNAi clones were generated from a sliding window analysis of matching 21-mers between the RNAi reagent and the *C. elegans* reference genome (ce10). We predicted no off-target sequence matches for the 29 clones used in our final analysis.

## Comparison of gene expression and embryonic lethality data

To test whether native gene expression of our target genes correlates with the embryonic lethality phenotypes, we downloaded microarray transcriptome data published by *Grishkevich et al. (2012)*. These data were collected on 4-cell embryos, which retain the maternally-inherited mRNA transcripts that were the targets of our study, and include three replicate values (following quantile normalization and $\log_{10}$ transformation) determined from three pools of 50 embryos each. We examined gene expression values for the 29 targeted genes, collected under control conditions, for five strains: AB2, CB4856, CB4857, N2, and RC301 (identical to PX174, which we tested in our study). We tested for the genotypic effect of strain with an ANOVA and for correlations between the transcriptome data and our estimates of gene-specific CGV using the Spearman Rank method. For each gene, we looked for correlation between the average gene expression value for each of the five strains and the strain coefficients from the strain-by-gene interaction term in our statistical analysis. We used the same generalized linear model structure as described above; in this analysis, we included 29 genes and five strains. We used a two-tailed binomial sign test to assess whether the 29 correlations were disproportionately positive or negative.

## Acknowledgements

We thank M Mana for RNAi clones, H-L Kao for help in computational data management, and P Cipriani, E Munarriz and K Erickson assistance with the high-throughput phenotyping platform. K Rattanakorn, C Karmel, and A Stavropoulos scored data images, which provided validation for Dev-StaR. We also thank L Noble for assisting in the analysis of whole-genome sequence data, D Pollard for useful discussions and advice, and three anonymous reviewers for their consideration of this manuscript. This work was supported by the NIH (grants GM089972, GM090557 and HD046236), the Zegar Family Foundation, and the Charles H Revson Foundation.

## Additional information

### Funding

| Funder | Grant reference number | Author |
| --- | --- | --- |
| Charles H. Revson Foundation | | Annalise B Paaby |
| National Institute of General Medical Sciences (NIGMS) | GM090557 | Annalise B Paaby |

| Zegar Family Foundation | | Kristin C Gunsalus<br>Fabio Piano<br>Matthew V Rockman |
|---|---|---|
| National Institutes of Health | HD046236 | Kristin C Gunsalus<br>Fabio Piano |
| National Institute of General<br>Medical Sciences (NIGMS) | GM089972 | Matthew V Rockman |

The funders had no role in study design, data collection and interpretation, or the decision to submit the work for publication.

### Author contributions

ABP, Conceived and designed experiments, Conducted the experiments, Analyzed the data and wrote the paper; AGW, Developed the image analysis algorithm DevStaR; DDR, Performed whole-genome sequencing on the worm strains; KCG, FP, Developed the phenotyping pipeline and the image analysis algorithm DevStaR; MVR, Conceived and designed experiments, Analyzed the data and wrote the paper

## Additional files

### Supplementary files

• Supplementary file 1. This Excel file reports correlations for strain-by-gene interaction coefficients for each pairwise combination of targeted genes. Above the diagonal are the estimated correlations; below the diagonal are the correlation p-values; in green are pairs of genes with interactions reported in the literature.

• Supplementary file 2. This extended table reports genome-wide SNPs associated with hatching phenotypes with p-values <0.0001 and <$1.64 \times 10^{-5}$ (*). The LD column indicates clusters of SNPs in strong disequilibrium with each other ($R^2 > 0.90$) across our test strains. A source data file has been deposited at Dryad under doi:10.5061/dryad.d5j06.

### Major datasets

The following datasets were generated:

| Author(s) | Year | Dataset title | Dataset URL | Database, license, and accessibility information |
|---|---|---|---|---|
| Paaby AB, White AG, Riccardi DD, Gunsalus KC, Piano F, Rockman MV | 2015 | Data from: Wild worm embryogenesis harbors ubiquitous polygenic modifier variation | http://dx.doi.org/10.5061/dryad.d5j06 | Available at Dryad Digital Repository under a CC0 Public Domain Dedication. |

The following previously published datasets were used:

| Author(s) | Year | Dataset title | Dataset URL | Database, license, and accessibility information |
|---|---|---|---|---|
| Andersen EC, Gerke JP, Shapiro JA, Crissman JR, Ghosh R, Bloom JS, Felix MA, Kruglyak L | 2015 | Data from: Chromosome-scale selective sweeps shape Caenorhabditis elegans genomic diversity | http://andersenlab.org/Research/Data/ | Publicly available. |

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
