## [Decision Letter]

Thank you for submitting your work entitled “Wild worm embryogenesis harbors
ubiquitous polygenic modifier variation” for peer review at *eLife*.
Your submission has been favorably evaluated by Diethard Tautz (Senior editor), a
Reviewing editor, and three reviewers, one of whom is a member of our Board of
Reviewing Editors.

The reviewers have discussed the reviews with one another and the Reviewing editor
has drafted this decision to help you prepare a revised submission.

This paper uses counts of dead embryos and living *C. elegans* larvae
as a phenotype to explore the extent of naturally occurring genetic variation in the
presence of mutations in embryonic genes. The authors report that knockdown of the
same gene has different effects in different isolates. These results represent a
systematic analysis of the background genetic effects that mitigate gene knockdown
on cell or developmental processes. The authors’ results have broad implications for
evolutionary, cellular and developmental biology.

Essential revisions:

1) The authors discovered that knocking down the same genes in different isolates has
different effects on embryonic lethality and the authors state that this observation
indicates that there is cryptic genetic variation. It is not clear that this
interpretation is correct. The alternative possibility is that the variation is not
“cryptic”: i.e. that development (and/or embryonic lethality) is really
phenotypically different in the different isolates.

The authors state that “Embryogenesis is an essential and stereotypic process.” While
it is certainly true that development is stereotypic within the standard lab strain,
N2, no study reports the extent of similarity of development across different
*C. elegans* isolates. Thus, it seems possible that development
could be different between different isolates. Similarly, spontaneous embryonic
lethality (in the absence of any gene knockdown) might also vary across *C.
elegans* isolates.

If development and spontaneous embryonic lethality are phenotypically different in
different isolates then it does not makes sense to say that a different genetic
basis for these traits is “cryptic”. Thus, the authors should either i) demonstrate
that development and/or embryonic lethality is phenotypically similar in the
different isolates, or ii) better justify in what sense the observed genetic
variation is “cryptic”, or iii) not refer to the observed genetic variation as
“cryptic”.

2) The authors state that “We have uncovered pervasive CGV among wild *C.
elegans* strains in the molecular and cellular processes of
embryogenesis” and their title is “Wild worm embryogenesis harbors ubiquitous
polygenic modifier variation”. However, the authors study embryonic lethality. While
their results might certainly have interesting implications for the genetic basis of
“cellular processes of embryogenesis” and other aspects of “embryogenesis”, strictly
speaking, their data does not address phenotypes other than embryonic lethality. The
paper would be improved by a more extensive discussion of the limitations of their
results in this regard.

3) The “non-informational” variants are of greatest interest to developmental and
evolutionary biologists. It is thus important to rule out as much of this as
possible. The authors themselves note that “gene-specific modifiers ... potentially
include gene-specific variation in RNAi sensitivity, perhaps due to heritable
variation in transcriptional licensing and variation in wild-type expression level
of the targeted gene, due to cis- or trans-acting regulatory variation.” This could
be tested by generating transcriptomes for five or so strains with unusually
divergent phenotype distributions. This would then allow the authors to directly
estimate the extent to which gene-by-strain interactions are due to expression
levels. RNAi knockdown efficacy is also an important informational variation to
explore, but much harder to address.

---

## [Author Response]

*1) The authors discovered that knocking down the same genes in different
isolates has different effects on embryonic lethality and the authors state that
this observation indicates that there is cryptic genetic variation. It is not
clear that this interpretation is correct. The alternative possibility is that
the variation is not “cryptic”: i.e. that development (and/or embryonic
lethality) is really phenotypically different in the different
isolates*.

[…]

*2) The authors state that “We have uncovered pervasive CGV among
wild* C. elegans *strains in the molecular and cellular processes
of embryogenesis” and their title is “Wild worm embryogenesis harbors ubiquitous
polygenic modifier variation”. However, the authors study embryonic lethality.
While their results might certainly have interesting implications for the
genetic basis of “cellular processes of embryogenesis” and other aspects of
“embryogenesis”, strictly speaking, their data does not address phenotypes other
than embryonic lethality. The paper would be improved by a more extensive
discussion of the limitations of their results in this regard*.

Points 1) and 2) identify important conceptual definitions in our work, and these
definitions are inter-related.

We define cryptic genetic variation in terms of a focal phenotype (in our case,
embryonic lethality). There is every expectation that the cryptic variation that
affects embryonic lethality also causes variation in cellular or developmental
phenotypes in a penetrant, non-cryptic manner, as the comment describes; we would
interpret such variation as a potential mechanism for the cryptic differences in
lethality. Wild isolates of *C. elegans* exhibit measureable
variation in several aspects of early development (15), but under standard conditions the embryos of
all strains hatch into larvae at rates approaching 100%. Variation in hatching rates
under standard conditions is radically amplified by RNAi perturbation, and this
newly exposed variation is CGV (Paaby & Rockman). Thus, to address comment 1),
we added the following paragraph to the Discussion, which expands upon our
conceptualization of cryptic variation and in doing so follows suggestions i) and
ii) above:

“We describe the variation we uncovered as “cryptic” because its effect on embryonic
survival is dramatically magnified under perturbed conditions. Without gene
perturbation, our strains exhibit little embryonic lethality. However, under
ordinary conditions the strains vary in gene expression and other cellular or
developmental phenotypes (15; 23),
which may be the mechanisms by which the cryptic alleles influence the penetrance of
the primary perturbation. Previously, we and others have described such differences
as variation in “intermediate” phenotypes (18; 43); whether a genetic variant is cryptic requires definition of the
focal phenotype, since even at the morphological level an allele can be cryptic in
one trait but penetrant in another (12).”

In defining the relationship between our embryonic lethality data and hypothetical
observations of cellular or developmental phenotypes, the added paragraph more
clearly delineates the extent of our results, which is the concern raised by point
2). We also agree with the reviewers that the language of “We have uncovered
pervasive CGV among wild *C. elegans* strains in the molecular and
cellular processes of embryogenesis” misstates the nature of our data, and we have
rephrased this sentence in the Discussion to “We have uncovered pervasive CGV among
wild *C. elegans* strains that modifies the probability that an
embryo will survive a gene perturbation.”

*3) The “non-informational” variants are of greatest interest to developmental
and evolutionary biologists. It is thus important to rule out as much of this as
possible. The authors themselves note that “gene-specific modifiers ...
potentially include gene-specific variation in RNAi sensitivity, perhaps due to
heritable variation in transcriptional licensing and variation in wild-type
expression level of the targeted gene, due to cis- or trans-acting regulatory
variation.” This could be tested by generating transcriptomes for five or so
strains with unusually divergent phenotype distributions. This would then allow
the authors to directly estimate the extent to which gene-by-strain interactions
are due to expression levels. RNAi knockdown efficacy is also an important
informational variation to explore, but much harder to address*.

This is a very important point and we are glad to expand upon our understanding of
the issue. We fully agree with the reviewers that experiments that evaluate
variation in efficacy of RNAi across strains would be very valuable. Such an
experiment would require extremely good estimates of transcript level (ideally
across developmental stage), both with and without RNAi. As suggested, transcript
data collected under standard conditions would also address whether and how the
strains vary in native gene expression, which is an informative aspect of
embryogenesis independent of the question of RNAi efficacy.

We also agree with the reviewers that the non-informational components of variation
are potentially the most interesting. Unfortunately, the myriad and complex ways in
which gene-specific knockdown might vary across strains, including by mechanisms
that are just beginning to be elucidated (e.g. transcriptional licensing), preclude
us from easily distinguishing between different classes of CGV mechanism. The review
points to one class of mechanism, variation in wild-type expression level, that we
can explicitly test. A very recently published paper (60) has shown that, across four strains with
perturbations to genes involved in the electron transport chain, a strain with a
more severe RNAi phenotype tends to have lower wild-type expression of the targeted
gene. We acquired embryonic transcriptome data from the work of [23] and tested for a
correlation between gene expression level and our estimates of embryonic lethality.
These results do not address whether variation in expression is, for example, via a
trans effect that elevates overall pathway activity and compensates for knockdown of
a particular member (i.e., the gene expression and RNAi phenotype share a common
cause), or a cis- or trans-acting effect specific to the targeted gene, revealing
pre-existing dosage variation that is tolerated under unperturbed conditions but
causal of the RNAi phenotype under perturbation (cf. [39]). Nevertheless, this analysis adds a
valuable component to our manuscript and a useful step forward in the larger
exploration of the problem.

Overall, we found that wild-type gene expression in our 29 genes, examined across
five strains, did not correlate with the embryonic lethality phenotypes of those
strains. We added the following to the Results section:

“Extragenic modifiers may work by affecting, in trans, the expression level of the
targeted gene. Recent work shows that differences in severity of RNAi phenotype, for
four *C. elegans* strains perturbed at electron transport chain
genes, are associated with differences in expression level of the targeted gene
(60). However, we find no
evidence for the reported pattern of lower expression explaining more severe
phenotypes. […] Although undetectable differences in transcript level may
nevertheless contribute to embryonic survival, these results suggest that much of
the gene-specific modifier effect we observe depends on variation beyond the target
gene.”

And we added the following to the Materials and methods (Comparison of gene
expression and embryonic lethality data):

“To test whether native gene expression of our target genes correlates with the
embryonic lethality phenotypes, we downloaded microarray transcriptome data
published by [23].
These data were collected on 4-cell embryos, which retain the maternally-inherited
mRNA transcripts that were the targets of our study, and include three replicate
values (following quantile normalization and log10 transformation) determined from
three pools of 50 embryos each. […]For each gene, we looked for correlation between
the average gene expression value for each of the five strains and the strain
coefficients from the strain-by-gene interaction term in our statistical analysis.
We used the same generalized linear model structure as described above; in this
analysis, we included 29 genes and five strains. We used a two-tailed binomial sign
test to assess whether the 29 correlations were disproportionately positive or
negative.”